# Enhanced Label Propagation through Affinity Matrix Fusion for Source-Free Domain Adaptation

## Abstract

Source-free domain adaptation (SFDA) has gained significant attention as a method to transfer knowledge from a pre-trained model on source domains toward target domains without accessing the source data. Recent research in SFDA has predominately adopted a self-training paradigm, focusing on utilizing local consistency constraints to refine pseudo-labels during self-training. These constraints encourage similar predictions among samples residing in local neighborhoods. Despite their effectiveness, the importance of global consistency is often overlooked. Moreover, such self-training-based adaptation processes suffer from the "confirmation bias": models use self-generated sub-optimal pseudo-labels to guide their subsequent training, resulting in a loop of self-reinforcing errors. In this study, we address the global consistency limitation by employing a label propagation method that seamlessly enforces both local and global consistency, leading to more coherent label predictions within the target domain. To mitigate the "confirmation bias", we propose utilizing an affinity matrix derived from current and historical models during the label propagation process. This approach takes advantage of different snapshots of the model to obtain a more accurate representation of the underlying graph structure, significantly enhancing the efficacy of label propagation and resulting in more refined pseudo-labels. Extensive experiments prove the superiority of our approach over the existing methods by a large margin. Our findings not only highlight the significance of incorporating global consistency within the SFDA framework but also offer a novel approach to mitigate the confirmation bias that arises from the use of noisy pseudo-labels in the self-training paradigm.

## 1 Introduction

Modern deep neural networks have achieved remarkable success in various applications. However, they are still susceptible to a fundamental challenge known as domain shift, where a model trained on data from one domain fails to generalize well to another domain with different statistical characteristics. To tackle this issue, unsupervised domain adaptation (UDA) (Ganin et al., 2016; Bousmalis et al., 2017; Hoffman et al., 2018) has emerged as a pivotal approach in the field of transfer learning. It allows models trained on a source domain with abundant labeled data to effectively adapt to a target domain where labeled data is scarce or unavailable.

Conventional works in UDA (Long et al., 2017; Yan et al., 2017; Cicek & Soatto, 2019) often focus on learning domain invariant features, assuming that the source data is available during the adaptation stage. However, this assumption may be infeasible in practice, primarily due to the sensitive nature of the source data. To address the practical limitations of traditional UDA methods, source-free domain adaptation (SFDA) (Kundu et al., 2020) is proposed as a promising direction to address the data privacy challenge while ensuring effective adaptation between domains.

Under the setting of SFDA, most methods rely on a self-training mechanism guided by pseudo-labels generated by the source model. However, with the distribution shift between the source and target domains, such pseudo-labels may be noisy and inaccurate. To tackle this issue, recent state-of-the-art SFDA methods (Yang et al., 2021a;b; 2022; Litrico et al., 2023) have adopted a

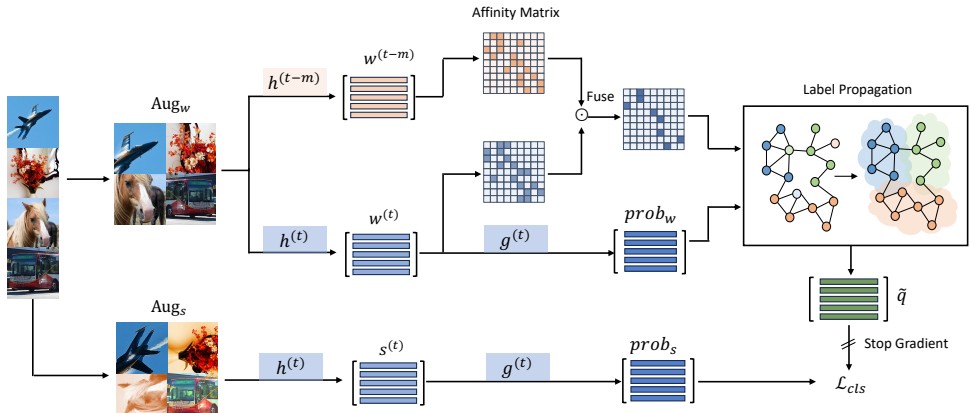

Figure 1: The framework of the proposed method. Classification predictions are initially generated using weakly augmented images of the target samples. These predictions undergo a subsequent refinement process through label propagation. The resulting refined pseudo-labels then serve as guiding information to train the classification model on strongly augmented samples.

neighborhood clustering approach for label information refinement, which promotes prediction consistency among local neighbors on the target domain. These approaches aggregate label information from nearby instances by incorporating local affinity in the target feature space to generate more reliable pseudo-labels. However, despite their effectiveness, these approaches fall short of fully exploiting the global structure of the target samples, and the label information formulated based on local clustering structures could still be susceptible to noise, leading to potential inaccuracies.

In this work, we propose a novel approach for SFDA by incorporating label propagation (Zhou et al., 2003a) to leverage both local and global relationships between the instances in the target domain. Label propagation refines the pseudo-labels by aligning the semantic relations of target samples with their similarities in the feature space. Different from neighborhood clustering (Yang et al., 2021a;b; 2022), our approach, based on label propagation, takes into account the broader structural dependencies beyond the immediate local neighborhood and achieves a more holistic understanding of the underlying relationships among the target instances. Moreover, we tackle the challenge of "confirmation bias" (Tarvainen & Valpola, 2017) in the self-training paradigm of SFDA. Pseudo-labels generated from models not optimally adapted to the target domain may contain noises, even after refinement. Using such noisy pseudo-labels to guide subsequent self-training may strengthen confidence in erroneous predictions, leading to a cycle of self-reinforcing errors. To mitigate the confirmation bias, we introduce an affinity matrix derived from both the current adapted model and historical models. Leveraging different snapshots of the models allows us to effectively correct confirmation bias and gain a more accurate representation of the relationships among target samples, thereby enhancing the accuracy and effectiveness of the label propagation process.

In addition, we investigate the critical role of data augmentation as a fundamental component in the self-training-based SFDA framework. By employing data augmentation on the target samples, we encourage the model to generate consistent outputs when subjected to input perturbations (Berthelot et al., 2019b; Xie et al., 2020; Sohn et al., 2020). The synergy between label propagation and data augmentation becomes apparent as they complement each other in the SFDA process. During the label refinement stage, label propagation enables us to obtain coherent and reliable pseudo-labels, guiding model adaptation. In the subsequent fine-tuning stage, data augmentation acts as a form of consistency regularization, enhancing the model's capability to address domain shift challenges.

## 2 RELATED WORK

**Domain Adaptation.** Standard Unsupervised Domain Adaptation (UDA) methods aim to adapt a model trained on a source domain to an unseen target domain, particularly in scenarios characterized by significant domain shift. Early UDA approaches predominantly relied on moment-matching

techniques to align the feature distributions across the source and target domains (Tzeng et al., 2014; Long et al., 2015; Sun et al., 2016). An alternative approach involves applying adversarial training on the features to learn domain-invariant representations (Tzeng et al., 2017; Zhang et al., 2019a; Cicek & Soatto, 2019). Recently, self-training has emerged as a prevailing trend within UDA, signifying a departure from traditional paradigms. Self-training-based methodology (Zou et al., 2018; Feng et al., 2021; Mei et al., 2020; Yu et al., 2021; Zhang et al., 2020) leverages labeled source data in conjunction with pseudo-labeled target data, simulating the conventions of standard supervised training. However, the success of these methodologies hinges upon access to both source and target data during the adaptation stage, which is often unrealistic due to concerns about data privacy associated with the source data.

**Source-Free Domain Adaptation.** With increasing concerns about data privacy, there has been growing interest in source-free domain adaptation. SFDA seeks to adapt models under scenarios where access to source data is prohibited. Two notable methods, TENT (Wang et al., 2020) and SHOT (Liang et al., 2020), employ entropy minimization and pseudo-labeling techniques to facilitate the adaptation of pre-trained source models to target data. Building upon this foundation, G-SFDA (Yang et al., 2021b), NRC (Yang et al., 2021a), AaD (Yang et al., 2022), and CoT (Wu et al., 2023) take a step further by refining pseudo-labels through the local consistency constraint. Multiple recent contributions (Huang et al., 2021; Chen et al., 2022; Litrico et al., 2023) introduce a novel approach by harnessing contrastive feature learning, inspired by MoCo (He et al., 2020) to exploit the unlabeled target sample to improve the robustness of the feature adaptation. Yi et al. (2023) introduces noise-robust training strategies to counter the influence of noisy pseudo-labels.

**Semi-Supervised Learning.** Consistency regularization is a widely adopted method in semi-supervised learning, which involves applying random perturbations to generate synthetic labels. These perturbations can take various forms, including dropout (Sajjadi et al., 2016), data augmentation (Berthelot et al., 2019a;b; Sohn et al., 2020), or adversarial perturbations (Miyato et al., 2016; 2018). Current state-of-the-art approaches combine both weak and strong augmentation techniques to maximize the benefits of augmentation consistency. Recent research highlights the importance of selecting suitable augmentation strategies (Xie et al., 2020), with RandAugment (Cubuk et al., 2020) and CTAugment (Berthelot et al., 2019a) emerging as popular choices. Typically, these methods generate hard pseudo-labels using weakly augmented images and then optimize the model's predictions on strongly augmented versions. In line with these approaches, our method combines the power of weak and strong data augmentation and demonstrates that data augmentation plays a critical role in self-training-based SFDA.

## 3 METHOD

The primary distinction between conventional Unsupervised Domain Adaptation (UDA) and Source-Free Domain Adaptation (SFDA) lies in the availability of source data. In SFDA, source data is completely absent during the model adaptation stage. In this context, the model adaptation relies solely on a pre-trained source model and the unlabeled target data $\{x_i\}_{i=1}^n$. We denote the source model as $f(\cdot) = g(h(\cdot))$, where $h(\cdot)$ represents the feature extractor and $g(\cdot)$ denotes a simple classifier comprising a single fully connected layer with softmax. This source model $f$ is trained in a supervised manner using the source data. We denote the feature embedding output from the feature extractor as $h(x) \in \mathbb{R}^d$. The classification probability predicted by the model is represented as $p(Y|x, g, h) = g(h(z)) \in \mathbb{R}^c$, with $c$ denoting the total number of classes. We use $\text{Aug}_s(\cdot)$ and $\text{Aug}_w(\cdot)$ to denote the strong and weak data augmentations on a given data point, respectively. For simplicity, we use $s_i$ to signify $h(\text{Aug}_s(x_i))$, and $w_i$ to represent $h(\text{Aug}_w(x_i))$.

**Overview**. We adopt a pseudo-label-based self-training framework for SFDA. Following the general structure of self-training paradigm, our model adaptation is guided by the following consistency loss:

$$\mathcal{L}_{cls} = \frac{1}{n} \sum_{i=1}^n l\Big(g(h(\text{Aug}_w(x_i))), g(h(\text{Aug}_s(x_i)))\Big) \tag{1}$$

$$= \frac{1}{n} \sum_{i=1}^n l\Big(g(w_i), g(s_i)\Big). \tag{2}$$

Here, the function $l$ represents the chosen loss function. $w_i$ and $s_i$ are the hidden representations for the weakly and strongly augmented $x_i$, respectively. Additionally, $g(w_i)$ is the pseudo-label generated based on $w_i$ and does not take gradient updates during self-training.

When a substantial domain gap exists between source and target data, prior research (Li et al., 2016; Nado et al., 2020) has shown that recalibrating the mean and variance statistics of batch normalization layers using target data is an efficient and cost-effective way to align the source model with the target data distribution. Building on these findings, our approach includes batch normalization adaptation during the initial stage of model adaptation, establishing a strong starting point for subsequent adaptation steps. Additional implementation details are available in Appendix A.1.

In the next phase, we employ a self-training paradigm to further adapt the source model to the target domain. To facilitate self-training, we generate pseudo-labels based on the source model after batch normalization adaptation. Recognizing that these labels may still contain noise due to domain shift, we leverage label propagation to refine them, and the refined pseudo-labels will be used to guide the subsequent self-training. Specifically, label propagation aims to align the relationships among target samples in the semantic space with those in the feature space. However, the affinity matrix describing the relationships of the target features may be inaccurate since the feature extractor may not be optimally aligned with the target domain. To capture more precise affinity relations among the target samples, we fuse the affinity matrix derived from the current and historical models. This strategic approach improves the accuracy of the affinity matrix, thereby enhancing the effectiveness of the label propagation process. The overall structure of our model is illustrated in Figure 1.

## 3.1 Label Propagation for Pseudo-Label Refinement

**Graph Construction**. Label propagation exploits the intrinsic similarity among target samples in the embedded feature space through graph construction, promoting smooth classification predictions in accordance with the graph's structural information. The affinity matrix, which depicts the relationships among target samples, is crucial for the effectiveness of label propagation. Relying solely on the current feature encoder $h^{(t)}$ to construct the graph's affinity matrix may introduce confirmation bias. To mitigate the confirmation bias, we propose a strategy incorporating both the current model $h^{(t)}$ and a historical model $h^{(t-m)}$ to create a more accurate affinity matrix.

To elaborate, we first employ the current adapted feature extractor $h^{(t)}$ to convert the target samples $\{x_i\}_{i=1}^n$ to the corresponding feature representations $\{w_i^{(t)}\}_{i=1}^n$, where $w_i^{(t)} = h^{(t)}(\text{Aug}_w(x_i))$. Then, we normalize these feature representations to unit vectors, $\{\tilde{w}_i^{(t)}\}_{i=1}^n$. From the normalized target features, we create the affinity matrix $A^{(t)} \in \mathbb{R}^{n \times n}$, where each element $a_{ij}^{(t)}$ signifies the similarity between the target instances $x_i$ and $x_j$ and is defined as:

$$a_{ij}^{(t)} = \begin{cases} [\langle \tilde{w}_i^{(t)}, \tilde{w}_j^{(t)} \rangle]_+^b, & \text{if } i \neq j \wedge \tilde{w}_j^{(t)} \in \mathcal{N}_K^{(t)}(i) \\ 0, & \text{otherwise} \end{cases} \tag{3}$$

where $\mathcal{N}_K^{(t)}(i)$ denotes the set of $K$ nearest neighbors of $\tilde{w}_i^{(t)}$, and $b$ is hyper parameter controlling the weight of the edges following Iscen et al. (2017) and is set to 3 in all experiments.

For each target sample $x_i$, the affinity matrix $A^{(t)}$ identifies its $K$ nearest neighbors based on the current model. However, since the feature encoder $h^{(t)}$ might not be optimally adapted to the target domain, the neighbors selected based solely on $h^{(t)}$ tend to be biased. To improve the accuracy of the affinity matrix, we consider an additional $K$ nearest neighbor affinity matrix $A^{(t-m)} \in \mathbb{R}^{n \times n}$ derived from the historical encoder $h^{(t-m)}$ (if $t < m$, we use $h^{(0)}$ as the historical encoder). If a particular sample $x_j$ is identified as one of the nearest neighbors by both the current and historical models, it is more likely to share the same label with the anchor sample $x_i$. This strategy helps mitigate potential bias from constructing the nearest neighbors graph solely based on the current model. We refine the affinity matrix by multiplying the similarity score derived from the current and historical models. The elements of resulting affinity matrix $A \in \mathbb{R}^{n \times n}$ are computed as follows:

$$a_{ij} = \begin{cases} a_{ij}^{(t)} \cdot a_{ij}^{(t-m)}, & \text{if } a_{ij}^{(t-m)} > 0 \\ a_{ij}^{(t)} \cdot \min_{j' \in \mathcal{N}_K^{(t-m)}(i)} a_{ij'}^{(t-m)}, & \text{otherwise.} \end{cases} \tag{4}$$

Subsequently, we update the affinity matrix to retain only the $k$ $(k < K)$ nearest neighbors of each target sample. For simplicity, we choose $k$ first and then set $K = k^2$ in all our experiments. By implementing this refinement procedure, we effectively account for the strengths of the affinity matrix derived from both the current and historical models, contributing to a more accurate representation of the relationships between target samples in the feature space.

Since the neighborhood relationship is not symmetric, we create a symmetric affinity matrix corresponding to an undirected graph by $W = A + A^\mathsf{T}$. We then apply normalization to obtain the normalized similarity matrix (Chung, 1997), denoted by $S = D^{-1/2}WD^{-1/2}$, where $D := diag(W\mathbf{1}_n)$ is the degree matrix and $\mathbf{1}_n$ is the n-dimensional constant one vector.

**Label Propagation**. Based on the graph constructed in the previous section, label propagation is used to refine the pseudo-labels of the target samples. Let $\mathcal{Q}$ denote the set of $n \times c$ matrices with non-negative entries. A matrix $Q = [Q_1^\mathsf{T}, \ldots, Q_n^\mathsf{T}]^\mathsf{T} \in \mathcal{Q}$ corresponds to the classification of the target samples with each row $Q_i^\mathsf{T}$ representing the classification probability for target instance $x_i$. Further, we define the initial label matrix $P \in \mathcal{Q}$ with the $i^{\text{th}}$ row corresponding to the initial pseudo-label $\hat{q}_i$ of the target sample $x_i$. To obtain the initial pseudo-label, we calculate the classification prediction using the current adapted model on the weakly augmented target sample, which is denoted as $p_i = g(w_i)$. The prediction $p_i$ then undergoes a sharpening process (Berthelot et al., 2019b) to yield the initial pseudo-label $\hat{q}_i$. Particularly, for any given probability distribution $p$, the sharpening process adjusts the distribution's "temperature" and is defined by the following operation:

$$\text{Sharpen}(p, T)_k := p_k^{\frac{1}{T}} \Big/ \sum_{k=1}^{c} p_k^{\frac{1}{T}}, \tag{5}$$

where $p$ represents a categorical distribution (in our approach, $p$ corresponds to the raw class prediction from the current model), and $T$ is the temperature hyperparameter, consistently set at $0.5$ for our experiments.

Starting from $P$, label propagation iteratively updates the labels according to the graph structure using the following formulation:

$$Q^{(\tau+1)} = \alpha S Q^{(\tau)} + (1 - \alpha)P, \tag{6}$$

where $Q^{(\tau)} \in \mathcal{Q}$ denotes the predicted label at progression step $\tau$, $S$ denotes the normalized weight matrix of the graph, and $\alpha \in (0, 1)$ controls the amount of propagated information. Note that the progression step is different from the training step, and we use label propagation to generate pseudo labels at the beginning of every training epoch. It has been shown in Zhou et al. (2003a;b) that the sequence $Q^{(\tau)}$ defined by Equation (6) converges to

$$Q^* = (I - \alpha S)^{-1}P, \tag{7}$$

where $I$ is an $n \times n$ identity matrix. In this equation, $Q^*$ represents the final refined pseudo-label matrix for the target samples. Each row of this matrix corresponds to the refined classification probability for the target sample $x_i$, which is denoted as $\tilde{q}_i$.

It is important to highlight the distinctions between the label propagation approach used in our work and label propagation algorithm used in semi-supervised learning (Zhou et al., 2003a; Iscen et al., 2019). In semi-supervised learning, label propagation aims to propagate information from labeled data to unlabeled data. The process involves diffusing the hard labels (one-hot encoded) from labeled samples to their neighboring unlabeled samples, which helps to provide pseudo-labels for the unlabeled data. In contrast, the label propagation approach in this work serves a different purpose. It focuses on leveraging the neighborhood structure to refine the noisy pseudo-labels obtained from the inadequately adapted model. Instead of using hard labels, our approach operates on soft pseudo-labels, which are the classification probabilities obtained from the current adapted model. These soft pseudo-labels offer richer information, providing the model's confidence level in its predictions for each target sample.

---

**Algorithm 1:** Source-Free Domain Adaptation with Label Propagation

---

**Data:** Source-pretrained model $f(\cdot) = g(h(\cdot))$ and target data $\mathcal{D}_t = \{x_i\}_{i=1}^n$
**Result:** Model adapted to the target domain

1 **for** $epoch = 1$ **to** $T$ **do**
2     **for** $i = 1$ **to** $n$ **do**
3         $w_i = h^{(t)}(x_i)$ ;                              // Extract target features
4         $p_i = g^{(t)}(w_i)$ ;                             // Generate raw predictions
5     **for** $(i,j) \in \{1, \cdots, n\}^2$ **do**
6         $a_{ij} \leftarrow$ affinity values using Equation 4
7     $W \leftarrow A + A^{\mathsf{T}}$ ;                 // Symmetrize the affinity matrix
8     $S \leftarrow D^{-1/2}WD^{-1/2}$ ;         // Normalize the affinity matrix
9     $Q^* \leftarrow (I - \alpha S)^{-1} P$ ;        // Compute the refined pseudo-labels
10     Update model $f$ with the loss function in Equation 10

---

## 3.2 SELF-LEARNING FOR MODEL ADAPTATION

Having obtained the refined pseudo-labels through the label propagation process, we can now leverage the refined pseudo-labels to supervise model adaptation on the target domain. Following (Iscen et al., 2019), we introduce an uncertainty weight, denoted as $\beta_i := 1 - \frac{H(\tilde{q}_i)}{\log(c)}$, to account for the uncertainty associated with the pseudo-labels, where $H : \mathbb{R}^c \to \mathbb{R}$ is the entropy function. Higher entropy values lead to lower weights, indicating higher uncertainty and reduced impact on the self-training stages, and vice versa. Hence, we formulate the loss function for model adaptation, which incorporates the uncertainty weights, as follows:

$$\mathcal{L}'_{cls} = \sum_{i=1}^n \beta_i \cdot l(\tilde{q}_i, g(s_i)). \tag{8}$$

Commonly used choices for $l$ are mean squared error loss (Berthelot et al., 2019b) or cross-entropy loss. In our work, following the approach of Yang et al. (2021a;b; 2022), we utilize the negative dot product loss, defined as, $l(q, p) = -q^{\mathsf{T}}p$. In Appendix A.2, we demonstrate the link between the employed loss function and the mean squared error loss.

In addition, to promote the diversity of the model's prediction on the target samples, we also adopt a prediction diversity loss as follows:

$$\mathcal{L}_{div} = \sum_{k=1}^c \bar{p}_k \log \bar{p}_k, \quad \bar{p} = \frac{1}{n}\sum_{i=1}^n g(s_i). \tag{9}$$

Subsequently, the overall training loss can be formulated as

$$\mathcal{L} = \mathcal{L}'_{cls} + \lambda \mathcal{L}_{div}, \tag{10}$$

where $\lambda$ is the hyper-parameter to balance the influence of the two loss terms.

## 3.3 ITERATIVE TRAINING

To summarize, we adopt an iterative self-training process that combines label propagation for pseudo-label refinement and self-training supervised by these refined pseudo-labels. The process begins with adapting the batch normalization layers using target domain data, providing a starting point for the subsequent iterative training. After the adaptation of batch normalization, the iterative training procedure unfolds as follows: First, we extract features and generate pseudo-labels for the target data using the current adapted model. Second, with the extracted feature, we construct a nearest neighbor-based graph and perform label propagation to refine the pseudo-labels for the target samples. Finally, we fine-tune the network for one epoch using the loss function defined in Equation 10. This iterative process is repeated for a total of $T$ epochs, facilitating gradual adaptation of the model to the target domain. In Algorithm 1, we provide a summary of the iterative self-training process.

# 4 EXPERIMENTS

We utilize three benchmark datasets in our experiments: Office-31 (Saenko et al., 2010), Office-Home (Venkateswara et al., 2017), and the VisDA-C (Peng et al., 2017) (details in Appendix A.3).

| Method | SF | plane | bcycl | bus | car | horse | knife | mcycl | person | plant | sktbrd | train | truck | Avg |
|---|---|---|---|---|---|---|---|---|---|---|---|---|---|---|
| CDAN (Long et al., 2018) | ✗ | 85.2 | 66.9 | 83.0 | 50.8 | 84.2 | 74.9 | 88.1 | 74.5 | 83.4 | 76.0 | 81.9 | 38.0 | 73.9 |
| SWD (Lee et al., 2019) | ✗ | 90.8 | 82.5 | 81.7 | 70.5 | 91.7 | 69.5 | 86.3 | 77.5 | 87.4 | 63.6 | 85.6 | 29.2 | 76.4 |
| MCC (Jin et al., 2020) | ✗ | 88.7 | 80.3 | 80.5 | 71.5 | 90.1 | 93.2 | 85.0 | 71.6 | 89.4 | 73.8 | 85.0 | 36.9 | 78.8 |
| STAR (Lu et al., 2020) | ✗ | 95.0 | 84.0 | 84.6 | 73.0 | 91.6 | 91.8 | 85.9 | 78.4 | 94.4 | 84.7 | 87.0 | 42.2 | 82.7 |
| RWOT (Xu et al., 2020) | ✗ | 95.1 | 80.3 | 83.7 | 90.0 | 92.4 | 68.0 | 92.5 | 82.2 | 87.9 | 78.4 | 90.4 | 68.2 | 84.0 |
| 3C-GAN (Li et al., 2020) | ✓ | 94.8 | 73.4 | 68.8 | 74.8 | 93.1 | 95.4 | 88.6 | 84.7 | 89.1 | 84.7 | 83.5 | 48.1 | 81.6 |
| SHOT (Liang et al., 2020) | ✓ | 94.3 | 88.5 | 80.1 | 57.3 | 93.1 | 94.9 | 80.7 | 80.3 | 91.5 | 89.1 | 86.3 | 58.2 | 82.9 |
| NRC (Yang et al., 2021a) | ✓ | 96.8 | 91.3 | 82.4 | 62.4 | 96.2 | 95.9 | 86.1 | 80.6 | 94.8 | 94.1 | 90.4 | 59.7 | 85.9 |
| HCL (Huang et al., 2021) | ✓ | 93.3 | 85.4 | 80.7 | 68.5 | 91.0 | 88.1 | 86.0 | 78.6 | 86.6 | 88.8 | 80.0 | **74.7** | 83.5 |
| AaD (Yang et al., 2022) | ✓ | 97.4 | 90.5 | 80.8 | 76.2 | 97.3 | 96.1 | 89.8 | 82.9 | 95.5 | 93.0 | 92.0 | 64.7 | 88.0 |
| C-SFDA (Karim et al., 2023) | ✓ | 97.6 | 88.8 | 86.1 | 72.2 | 97.2 | 94.4 | 92.1 | 84.7 | 93.0 | 90.7 | 93.1 | 63.5 | 87.8 |
| CtO (Wu et al., 2023) | ✓ | 98.2 | 91.0 | 86.4 | **78.0** | 97.6 | **98.8** | 91.8 | 84.8 | **96.6** | 94.7 | 93.7 | 53.3 | 88.7 |
| **Ours** | ✓ | **98.4** | **91.9** | **87.8** | 68.9 | **97.8** | 97.7 | **93.3** | **85.8** | 96.5 | **96.8** | **94.2** | 65.6 | **89.6** |

Table 1: Accuracies (%) on VisDA-C (Synthesis → Real) for ResNet101-based methods. The column "SF" denotes whether the setting is source-free or not.

| Method | SF | Ar→Cl | Ar→Pr | Ar→Rw | Cl→Ar | Cl→Pr | Cl→Rw | Pr→Ar | Pr→Cl | Pr→Rw | Rw→Ar | Rw→Cl | Rw→Pr | Avg |
|---|---|---|---|---|---|---|---|---|---|---|---|---|---|---|
| MCD (Saito et al., 2018) | ✗ | 48.9 | 68.3 | 74.6 | 61.3 | 67.6 | 68.8 | 57.0 | 47.1 | 75.1 | 69.1 | 52.2 | 79.6 | 64.1 |
| CDAN (Long et al., 2018) | ✗ | 50.7 | 70.6 | 76.0 | 57.6 | 70.0 | 70.0 | 57.4 | 50.9 | 77.3 | 70.9 | 56.7 | 81.6 | 65.8 |
| SAFN (Xu et al., 2019) | ✗ | 52.0 | 71.7 | 76.3 | 64.2 | 69.9 | 71.9 | 63.7 | 51.4 | 77.1 | 70.9 | 57.1 | 81.5 | 67.3 |
| MDD (Zhang et al., 2019b) | ✗ | 54.9 | 73.7 | 77.8 | 60.0 | 71.4 | 71.8 | 61.2 | 53.6 | 78.1 | 72.5 | 60.2 | 82.3 | 68.1 |
| BNM (Cui et al., 2020) | ✗ | 52.3 | 73.9 | 80.0 | 63.3 | 72.9 | 74.9 | 61.7 | 49.5 | 79.7 | 70.5 | 53.6 | 82.2 | 67.9 |
| SRDC (Tang et al., 2020) | ✗ | 52.3 | 76.3 | 81.0 | 69.5 | 76.2 | 78.0 | 68.7 | 53.8 | 81.7 | 76.3 | 57.1 | 85.0 | 71.3 |
| SHOT (Liang et al., 2020) | ✓ | 57.1 | 78.1 | 81.5 | 68.0 | 78.2 | 78.1 | 67.4 | 54.9 | 82.2 | 73.3 | 58.8 | 84.3 | 71.8 |
| NRC (Yang et al., 2021a) | ✓ | 57.7 | 80.3 | 82.0 | 68.1 | 79.8 | 78.6 | 65.3 | 56.4 | 83.0 | 71.0 | 58.6 | 85.6 | 72.2 |
| HCL (Huang et al., 2021) | ✓ | **64.0** | 78.6 | 82.4 | 64.5 | 73.1 | **80.1** | 64.8 | **59.8** | 75.3 | **78.1** | **69.3** | 81.5 | 72.6 |
| AaD (Yang et al., 2022) | ✓ | 59.3 | 79.3 | 82.1 | 68.9 | 79.8 | 79.5 | 67.2 | 57.4 | 83.1 | 72.1 | 58.5 | 85.4 | 72.7 |
| C-SFDA (Karim et al., 2023) | ✓ | 60.3 | 80.2 | **82.9** | **69.3** | 80.1 | 78.8 | 67.3 | 58.1 | **83.4** | 73.6 | 61.3 | 86.3 | **73.5** |
| **Ours** | ✓ | 62.9 | **81.1** | 82.2 | 68.2 | **80.9** | 78.3 | **67.4** | 57.5 | 82.0 | 73.6 | 61.7 | **86.5** | **73.5** |

Table 2: Accuracies (%) on Office-Home for ResNet50-based methods.

## 4.1 EXPERIMENT RESULTS

We comprehensively evaluate our proposed method against the current state-of-the-art approaches across a wide range of domain adaptation benchmark datasets, and the results are summarized in Tables 1–3. Notably, on the most challenging and largest dataset, VisDA, our method surpasses the existing SFDA methods by a large margin (1.6% compared to AaD). Furthermore, on both Office-Home and Office-31 datasets, our method achieves results that are comparable to the more complex C-SFDA method, which generates the pseudo-labels based on 12 augmented copies of the image. On the contrary, our approach generates pseudo-labels based on a single copy of a weakly augmented image, resulting in significantly lower computational costs compared to C-SFDA.

## 4.2 ANALYSIS

**Batch Normalization Adaptation.** Batch normalization adaptation is particularly effective when there is a substantial domain gap between the source and target domains, as is the case with VisDA. In Figure 2, we illustrate the benefit of batch normalization adaptation using the VisDA dataset. We randomly selected 400 samples from each class in the target domain and generated t-SNE plots before and after applying batch normalization adaptation. The plots clearly demonstrate that the model becomes more aligned with the target samples after adaptation, resulting in distinct clustering patterns among samples of the same class. Moreover, as shown in Table 4, the application of batch normalization adaptation alone substantially improves the accuracy of the source model. It increases the accuracy from 46.1% to 66% on VisDA dataset and from 59.6% to 60.7% on Office-Home dataset. This initial adaptation step sets a resilient foundation for subsequent adaptation phases.

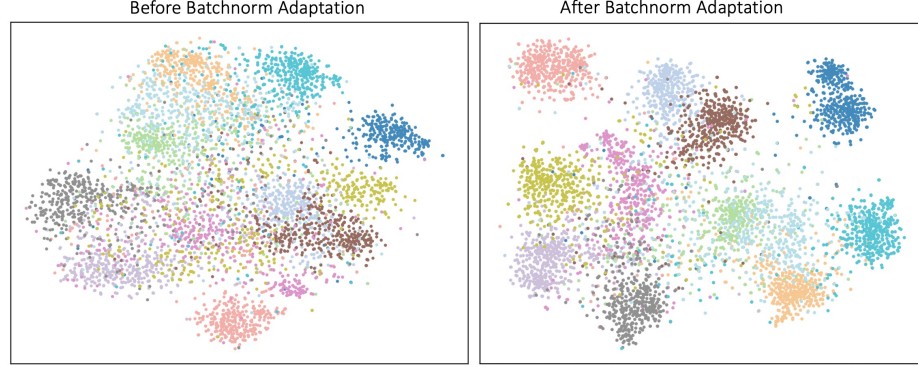

Figure 2: Visualizing batch normalization adaptation effects using VisDA dataset: T-SNE plots show 400 randomly selected samples per class from the VisDA dataset before (left) and after (right) batch normalization adaptation.

| Method | SF | A→D | A→W | D→W | W→D | D→A | W→A | **Avg** |
|---|---|---|---|---|---|---|---|---|
| MCD (Saito et al., 2018) | ✗ | 92.2 | 88.6 | 98.5 | 100.0 | 69.5 | 69.7 | 86.5 |
| CDAN (Long et al., 2018) | ✗ | 92.9 | 94.1 | 98.6 | 100.0 | 71.0 | 69.3 | 87.7 |
| MDD (Zhang et al., 2019b) | ✗ | 90.4 | 90.4 | 98.7 | 99.9 | 75.0 | 73.7 | 88.0 |
| DMRL (Wu et al., 2020) | ✗ | 93.4 | 90.8 | 99.0 | 100.0 | 73.0 | 71.2 | 87.9 |
| MCC (Jin et al., 2020) | ✗ | 95.6 | 95.4 | 98.6 | 100.0 | 72.6 | 73.9 | 89.4 |
| SRDC Tang et al. (2020) | ✗ | 95.8 | 95.7 | 99.2 | 100.0 | 76.7 | 77.1 | 90.8 |
| SHOT (Liang et al., 2020) | ✓ | 94.0 | 90.1 | 98.4 | 99.9 | 74.7 | 74.3 | 88.6 |
| 3C-GAN Li et al. (2020) | ✓ | 92.7 | 93.7 | 98.5 | 99.8 | 75.3 | 77.8 | 89.6 |
| NRC (Yang et al., 2021a) | ✓ | 96.0 | 90.8 | 99.0 | **100.0** | 75.3 | 75.0 | 89.4 |
| HCL (Huang et al., 2021) | ✓ | 94.7 | 92.5 | 98.2 | **100.0** | 75.9 | 77.7 | 89.8 |
| AaD (Yang et al., 2022) | ✓ | **96.4** | 92.1 | **99.1** | 100.0 | 75.0 | 76.5 | 89.9 |
| C-SFDA (Karim et al., 2023) | ✓ | 96.2 | **93.9** | 98.8 | 99.7 | **77.3** | 77.9 | **90.5** |
| **Ours** | ✓ | 94.2 | 93.3 | 98.5 | 99.9 | **77.3** | **78.2** | 90.3 |

Table 3: Accuracies (%) on Office31 for ResNet50-based methods.

**Data Augmentation.** In Table 4, we also conduct an ablation study of the remaining components of the model on VisDA and Office-Home. Our approach incorporates two strong augmentations, $\text{Aug}_s$ and $\text{Aug}'_s$, during the self-training stage, with the detailed information provided in Appendix A.4. The result in Table 4 highlights the importance of data augmentation, particularly for VisDA, where the target data contains objects of vastly different scales. In comparison to the baseline, which employs only standard weak augmentation during the self-training stage, the introduction of $\text{Aug}_s$ and $\text{Aug}'_s$ during self-training leads to an approximate $2.4\%$ increase in model accuracy on VisDA and an improvement of approximately 0.5% on Office-Home.

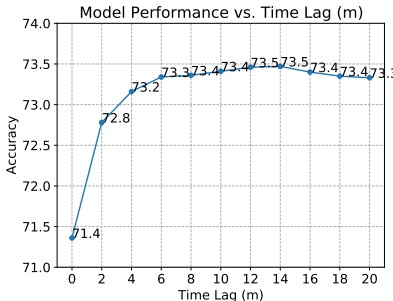

Figure 3: Model performance across various time lag $(m)$ settings in historical affinity fusion. Ablation study conducted on Office-Home dataset, evaluating the accuracy across all 12 tasks.

| Method | Avg Acc. | |
|---|---|---|
| | VisDA | Office-Home |
| Source model | 46.1 | 59.6 |
| Batchnorm adaptation | 66.0 | 60.7 |
| Baseline (ours) | 86.5 | 70.9 |
| Baseline + HAF | 87.4 | 72.9 |
| Baseline + $\text{Aug}_s$ | 87.9 | 71.2 |
| Baseline + $\text{Aug}_s$ + $\text{Aug}'_s$ | 88.9 | 71.4 |
| Baseline + $\text{Aug}_s$ + $\text{Aug}'_s$ + HAF | 89.6 | 73.5 |

Table 4: Ablation study of different modules. "Baseline" denotes our approach using only standard weak augmentation in the self-training stage and without utilizing affinity fusion during the label propagation stage. "HAF" stands for historical affinity fusion.

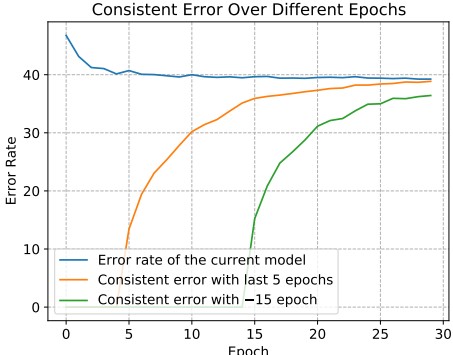 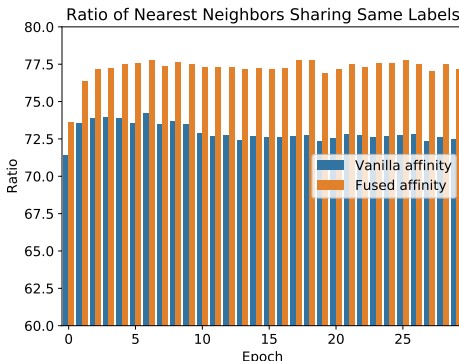

Figure 4: **Left Panel**: the percentage of predictions that are both incorrect in the current model and consistent with predictions made by historical models. **Right Panel**: the ratio of connected samples sharing the same label in the vanilla affinity matrix as well as the fused affinity matrix. Note that the affinity matrix at time 0 is fused with itself.

**Label Propagation.** In Figure 4, we conduct an ablation study on the label propagation algorithm used for pseudo-label refinement. As an illustrative example, we focus on adaptation from Artistic images (Ar) to Clipart images (Cl) within the Office-Home dataset.

In the left panel of Figure 4, we depict the cumulative effect of confirmation bias during the model adaptation stage. This is revealed through two key metrics: the prediction error rate of the current adapted model and the rate of predictions that are both incorrect and consistent with historical models from the preceding 5 epochs. As training progresses, the model tends to repeat the same errors, with almost all incorrect predictions aligning with those from prior models. This phenomenon occurs because noisy pseudo-labels are used to guide model adaptation, leading the model to reinforce these label inaccuracies. Then, we examine the agreement between the historical model $f^{(t-m)}$ with $m = 15$ and the current model $f^{(t)}$. Notably, the proportion of predictions that are simultaneously incorrect in both $f^{(t)}$ and $f^{(t-m)}$ is significantly lower compared to that of the immediate historical model. By integrating information from historical models that significantly differ from the current model, we effectively rectify a portion of the errors present in the current model.

To assess the efficacy of historical affinity fusion, we inspect the accuracy of the fused affinity matrix compared to the vanilla affinity matrix. This assessment involved analyzing the ratio of connected samples (non-zero entries in the affinity matrix) that share the same ground truth label throughout the training process, as depicted in the right panel of Figure 4. Remarkably, the fused affinity matrix consistently demonstrated higher accuracy than the vanilla affinity matrix, which subsequently improved the efficacy of label propagation for pseudo-label refinement.

In Figure 3, we delve into the impact of the hyper-parameter $m$ associated with affinity matrix fusion. This parameter signifies the time lag between the current model and the historical model, with $m = 0$ corresponding to the baseline scenario where label propagation relies solely on the vanilla affinity matrix. As depicted in the plot, fusion of historical affinity matrices substantially enhances the model's performance. Moreover, the model's performance remains stable across different values of $m$ once it exceeds 5.

## 5 CONCLUSION

This study introduces a novel approach for Source-Free Domain Adaptation (SFDA) that harnesses the combined power of label propagation and a self-training framework. Our approach builds upon prior methods by leveraging local and global relationships within the target samples through label propagation. To further enhance the accuracy of label propagation, we refine the reliability of the affinity matrix by fusing the affinity matrix derived from both current and historical models. We also incorporate data augmentation to improve the model's robustness against perturbations. Experimental results across various domain adaptation benchmarks affirm the efficacy of our method.

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

# A APPENDIX

## A.1 BATCH NORMALIZATION ADAPTATION

For datasets exhibiting significant domain shifts, the statistical disparity in mean and variance between source and target data is a crucial challenge hindering domain adaptation, leading to performance degradation when directly applying the source model to the target domain. A straightforward remedy is using target domain data to recalibrate statistics in batch normalization layers.

Batch normalization (Ioffe & Szegedy, 2015) first standardizes features within mini-batches and subsequently applies an affine transformation. The process can be mathematically expressed as: $\hat{z}_j = \frac{z_j - \mu_j}{\sigma_j}\gamma_j + \beta_j$, where $z_j$ and $\hat{z}_j$ represent the input and output activation of the $j^{th}$ neuron in a layer. $\mu_j$ and $\sigma_j$ are the corresponding mean and standard deviation calculated based on the mini-batches, while $\gamma_j$ and $\beta_j$ are parameters optimized using the loss function.

To align the source model with the distribution of the target domain, we update the statistics of the running mean and variance by processing all target data samples sequentially. Despite its simplicity, this process effectively aligns the feature extractor with the distribution of the target dataset, which provides a strong starting point for the subsequent adaptation stage.

## A.2 DISCUSSION ABOUT THE LOSS FUNCTION

In the context of self-training frameworks, commonly used loss functions include mean squared error loss Berthelot et al. (2019b) and cross-entropy loss. In our work, we adopt the dot product loss function $l(q, p) = -q^\intercal p$ , following Yang et al. (2021a;b; 2022). The dot product loss between the refined pseudo-label $\tilde{q}$ and the model prediction $p$ can be expressed as:

$$l(\tilde{q}, p) = -\tilde{q}^\intercal p = \frac{1}{2}(\|\tilde{q} - p\|^2 - \|\tilde{q}\|^2 - \|p\|^2) \tag{11}$$

Since the pseudo-label $\tilde{q}$ does not undergo gradient flow and can be treated as constant, this loss function is equivalent to:

$$l(\tilde{q}, p) = \|\tilde{q} - p\|^2 - \|p\|^2. \tag{12}$$

The first component in the above loss function represents the mean squared error (MSE) loss between the pseudo-label $\tilde{q}$ and the model prediction $p$. And the second term, $-\|p\|^2 = \sum_{k=1}^c p_k^2$, signifies the concept of Collision entropy. Notably, $\sum_{k=1}^c p_k^2 \leq \sum_{k=1}^c p_k = 1$, implying that the second term is lower-bounded by $-1$. This minimum is reached when $p_k^2 = p_k$ holds for all $k$, indicating that $p$ is a one-hot key vector. Hence, similar to Shannon Entropy $H(p) = -\sum_{k=1}^c p_k \log(p_k)$, $-\|p\|^2$ also attains its minimum when $p$ represents a one-hot key probability distribution and minimizing $-\|p\|^2$ amplifies the model's prediction discriminability. Hence, the loss function in Equation 12 encompasses two objectives: aligning the model prediction with the pseudo-labels and enhancing prediction discriminability.

## A.3 DATASET

**VisDA-C dataset**. The VisDA-C dataset (Peng et al., 2017) is a large-scale synthetic-to-real dataset presenting two extremely distinct domains: synthetic images and real images. Each domain consists of images from 12 classes. The synthetic domain contains 150K rendered 3D images captured from various poses and lighting conditions. The real domain comprises around 55K real-world images sourced from the Microsoft COCO dataset (Lin et al., 2014).

**Office-Home dataset**. Office-Home (Venkateswara et al., 2017) is a complex and challenging UDA benchmark dataset. It encompasses four visually diverse domains: Artistic images (**Ar**), Clipart images (**Cl**), Product images (**Pr**), and Realworld images (**Rw**). This dataset features 12 transfer learning tasks and contains a total number of 15,500 images from 65 classes.

**Office-31 dataset**. Office-31 (Saenko et al., 2010) serves as a small-scale benchmark with three domains, Amazon (**A**), DSLR (**D**), and Webcam (**W**). A total of 4,110 images are distributed across 31 categories, reflecting real-world scenarios.

## A.4 IMPLEMENTATION DETAILS

Our implementation utilizes standard classification architectures comprising a feature extractor followed by a classifier. To ensure direct comparison with related methods, we adopt the ResNet50 (He et al., 2016) backbone for the Office-31 and Office-Home datasets, and the ResNet101 (He et al., 2016) backbone for the VisDA dataset. Following SHOT (Liang et al., 2020), we introduce an additional 256-dimensional fully connected layer and an additional batch normalization layer to the CNN encoder. We further apply weight normalization (Salimans & Kingma, 2016) to the final classification layer. For source training, we initialize the models with ImageNet-1K (Deng et al., 2009) pre-trained weights. During this phase, the source model is trained with the standard cross-entropy loss with label smoothing regularization (Szegedy et al., 2016). When adapting to the target domain, we adopt SGD with a momentum of 0.9 and a batch size of 64 for all datasets. The learning rate for the CNN encoder is set to half of that for the bottleneck and classifier. Specifically, a learning rate of 0.01 is employed with a step decay scheduler in place. We train the model for 30 epochs on Office-31 and Office-Home, and 20 epochs for VisDA.

During implementation, three critical hyperparameters are considered: $k$, denoting the number of nearest neighbors; $m$, representing the time lag between the current and historical models; and $\lambda$, which signifies the weight of the diversity loss. Similar to previous local clustering-based approaches, we set $k$ to 3 for Office-Home and Office-31, and to 5 for VisDA. For $m$, we set it to 10 for VisDA and Office-31, and 15 for Office-Home. Regarding $\lambda$, we assign it a value of 0.1 for Office-Home and Office-31. However, due to the highly unbalanced nature of the target data in VisDA, we have observed that setting $\lambda$ to 0 (i.e., excluding the diversity loss) yields the best performance on the target domain.

Our approach for SFDA involves one weak augmentation $\text{Aug}_w$, and two strong augmentations $\text{Aug}_s$ and $\text{Aug}_s'$. We employ the standard crop-and-flip technique for weak augmentation, which is used to generate pseudo-labels. For strong augmentations, we adopt the augmentation strategy in MoCo V2 (Chen et al., 2020) as $\text{Aug}_s$, which comprises random resized crop, random color jittering, grayscale conversion, and Gaussian blur. In the case of $\text{Aug}_s'$, we use the same augmentation suite as MoCo V2 while excluding the randomly resized crop augmentation.

## A.5 DISCUSSION OF SOME RELEVANT METHODS

In this section, we compare our approach with some existing works closely related to ours.

Recent work by Zhang et al. (Zhang et al., 2020) focuses on unsupervised domain adaptation and integrates label propagation within a self-training framework, sharing certain similarities with our approach. Notably, it utilizes both labeled source data and unlabeled target data for label propagation. However, under Source-Free Domain Adaptation (SFDA), where only target data is available, the affinity matrix is inherently noisier due to the model's lack of adaptation to the target domain. In response, our method strategically leverages both current and historical models to construct a more precise affinity matrix, a critical factor for the success of label propagation under SFDA conditions.

HCL (Huang et al., 2021) also leverages historical models, similar to our approach. HCL relies on historical models to build positive and negative pairs for contrastive learning and gauges the confidence of pseudo-labels based on the agreement between current and historical models, without a specific refinement process. In contrast, our method utilizes label propagation for pseudo-label refinement, a crucial step that significantly enhances the effectiveness of the adaptation process. Moreover, our approach goes beyond a simple reliance on historical models; we specifically leverage them to construct a more precise nearest-neighbor graph. This strategic incorporation is integral to label propagation's effectiveness, highlighting historical models' nuanced contributions to our method's performance.

CtO (Wu et al., 2023) is connected to label propagation, utilizing the labels of immediate neighbors as a supervision signal for each sample, a strategy similar to NRC (Yang et al., 2021a) and AaD (Yang et al., 2022). However, as mentioned in the main paper, refining pseudo-labels solely

based on nearest neighbors falls short of effectively exploiting the global structure of target samples, resulting in suboptimal outcomes. In contrast, our approach leverages both local and global structures to refine pseudo-labels. Another key contribution involves incorporating the historical model into the label propagation process. While leveraging historical models is not new in semi-supervised learning, directly ensembling them for pseudo-label refinement in SFDA tends to yield suboptimal results. Motivated by the observation that similar samples form clusters in the feature space using both current and historical models, we combine both to construct a more accurate affinity matrix. This integration proves crucial for enhancing label propagation effectiveness and, consequently, improving overall performance.

## A.6 ABLATION FOR THE HYPERPARAMETERS

Our approach integrates several established techniques, including batch normalization adaptation, uncertainty weight, and diversity loss. In Table 5, we examine the impact of these hyperparameters on model performance, yielding the following observations: 1) Batch normalization adaptation proves to be more beneficial for VisDA, where a larger domain gap exists between the source and target domains. 2) The inclusion of uncertainty weight leads to a sizable improvement in model performance across all datasets, indicating its positive influence on the learning process. 3) Diversity loss contributes to model improvement for both Office-Home and Office 31. However, it exhibits a negative impact on VisDA, primarily due to the dataset's inherent class imbalance.

| Method | Avg Acc. | | |
|---|---|---|---|
| | VisDA | Office-Home | Office31 |
| **Ours** | 89.6 | 73.5 | 90.3 |
| Ours w/o batch norm adaptation | 88.1 | 73.3 | 90.1 |
| Ours w/o uncertainty weight | 88.9 | 73.2 | 88.5 |
| Ours w/o diversity loss | $88.3^*$ | 72.0 | 89.9 |

Table 5: Ablation study of various modules across all three datasets. Our approach for VisDA did not incorporate diversity loss, as explained in Section A.4. The results denoted with $*$ represent the performance of our approach with diversity loss ($\lambda = 0.1$) on VisDA.

