# OpenReview forum: "Enhanced Label Propagation through Affinity Matrix Fusion for Source-Free Domain Adaptation"
_ICLR.cc/2024/Conference — Submitted to ICLR 2024_

### Official Review · Reviewer_CpfT · 2023-10-16

**Soundness:** 3 good
**Presentation:** 3 good
**Contribution:** 2 fair
**Rating:** 6
**Confidence:** 4

**Summary:**

This paper addresses source-free domain adaptation (SFDA) task, where a source pretrained model is adapted to an unlabeled target domain in the absence of the source data. The submission proposes to tackle the problem by label propagation, along with considering historical model embedding and introducing data augmentations.

**Strengths:**

- Motivation is clear and sound: The proposed method is based on label propagation, to deal with the noisy pseudo labels/predictions, historical feature embeddings are considered in the affinity matrix definition.

- Experiments on several benchmarks show the effectiveness of the proposed method.

**Weaknesses:**

- Label propagation is already investigated in the normal domain adaptation [a] (and also source-free domain adaptation [b]), the authors may consider discussing those works in the related works section.


- One major improvement comes from data augmentations, which is a universal way to improve the performance in unsupervised learning. I think in the ablation study, results with Baseline + HAF should be provided on all datasets, to indicate how much improvement data augmentation could introduce into label propagation.

- (Major) Some techniques used in the submission are quite general and not new, such as BN statistics adaptation (which is achieved by forwarding target data to source model before starting adaptation), and uncertainty weight in Eq. 8 (and also diversity loss in Eq. 9). As there is no ablation study about those techniques, I posit they can improve all existing SFDA methods. For fair comparison, at least results without BN adaptation should be provided. Or BN adaptation should be added to all existing methods in the main tables, otherwise we do not know how many gains are really from label propagation, which is the key module in the submission.

- In the appendix, it mentions that there are two strong augmentations deployed, how about only using one or multiple ones in turn?

- The hyperparameter $\lambda$ is set manually (0.1 for office datasets and 0 for visda), how about the performance without using diversity loss, or using a decaying weight on this term?

***Reference:***

[a] Label Propagation with Augmented Anchors: A Simple Semi-Supervised Learning baseline for Unsupervised Domain Adaptation. ECCV 2020

[b] Chaos to Order: A Label Propagation Perspective on Source-Free Domain Adaptation. ACM MM 2023

**Questions:**

Please check the weakness above, without detailed ablation studies, currently I think the submission is a bit incremental by combining several general techniques.

By the way, I think the method is also somehow similar to BYOL. The label propagation part with weak augmentation could be regarded as the projection layer in BYOL, though the operation in the submission is in the output space (classification prediction). The authors could think about it.

---

> ### Author Response · Authors · 2023-11-17
> **Rebuttal**
>
> Thank you for your time and suggestive comments.  We will improve the paper to include the discussions below.
>
> **W1.**  We have incorporated the suggested papers into the discussion of the related works in Section A.5. We want to highlight that A$^2$LP relies on both labeled source data and unlabeled target data for label propagation. In the context of SFDA, where only target data is available, the affinity matrix is inherently noisier due to the model's lack of adaptation to the target domain. We address this challenge by exploiting both current and historical models to construct a more precise affinity matrix—a crucial factor for the success of label propagation, especially under SFDA conditions.
>
> On the other hand, CtO [b] used a local clustering approach instead of label propagation (even though the title contains label propagation), which is similar to NRC and AaD. However, as noted in the main paper, refining pseudo-labels solely based on nearest neighbors fails to effectively exploit the global structure of target samples, resulting in suboptimal outcomes.
>
> **W2.** Thank you for the suggestion. We have provided the results for Baseline + HAF without strong data augmentation in the ablation study, and the detailed outcomes are presented in Table 4.
>
> **W3.** Our approach has integrated several established techniques, including BN adaptation, uncertainty weight, and diversity loss. While these techniques are not the primary focus of this paper, we acknowledge the importance of understanding their impact. We have included an ablation study in Appendix A.5 to address this concern.
>
> In addition, we also check the effectiveness of BN adaptation when integrated with existing methods, as summarized in the table below.  Please note that the code for some baseline models, such as CoT and C-SFDA, is not publicly available. Instead, we provided the results for AaD and NRC, which are the best-performing models with accessible code. Note that AaD and NRC do not benefit significantly from BN, and our method maintains strong performance even without BN.
> The integration of BN adaptation is particularly synergistic with our approach, as it helps generate more accurate similarity measures and affinity matrices. Since our label propagation process relies on the affinity matrix for pseudo-label refinement, the improved accuracy facilitated by BN adaptation significantly improves the effectiveness of our method. In addition, incorporating BN adaptation into SFDA is also a novel contribution in our paper.
>
> | Method            |Avg Acc. on VisDA|
> | --------             | -------    |
> | NRC                            |    85.9   |
> | NRC + BN adaptation |   86.0    |
> | AaD                             |   87.9    |
> | AaD + BN adaptation  |   87.7    |
> | Ours w/o BN adaptation  |   88.1    |
> | Ours w/ adaptation          |   89.6    |
>
>
> **W4.** We have included an ablation study on the impact of using a single strong augmentation in Table 4 in the main paper.  Additionally, the outcomes of incorporating three augmentations are included in the table below. The results indicate that integrating more strong augmentations during the self-training stage yields only marginal improvements. For efficiency considerations, we opt to use two strong augmentations.
>
> | Method                |Avg Acc. on VisDA|
> | --------                 | -------    |
> | Ours  w/   2 Aug$_s$     |    89.6  |
> | Ours w/ 3 Aug$_s$        |   89.8    |
>
>
> **W5.** The details regarding hyperparameters are outlined in Section A.6. Notably, incorporating diversity loss for VisDA adversely affects model performance. Additionally, the removal of diversity loss has a more pronounced impact on the Office-Home dataset while causing only minor changes in performance for Office31.
>
>
> **Q2.** Thank you for the suggestion! Indeed, the self-training framework in our approach shares some similarities with BYOL. As you mentioned, there is some distinction as we focus on the prediction space. We will investigate this approach in our future study on SFDA.

---

> > ### Comment · Reviewer_CpfT · 2023-11-19
> >
> > Thanks to the authors for the response. Although absorbing many existing works (which means not quite novel), overall this submission is a reasonable method working well, and every module is well motivated and clear, I improve my score from 5 to 6.

---

### Official Review · Reviewer_113e · 2023-11-01

**Soundness:** 2 fair
**Presentation:** 2 fair
**Contribution:** 2 fair
**Rating:** 5
**Confidence:** 3

**Summary:**

This paper proposes to leverage label propagation and self-supervised learning methods in SFDA (Source-Free Domain Adaptation) problem, aiming to obtain the global semantic information and reduce the confirmation bias in the target domain. Specifically, the authors use both historical model snapshots and the current model to build the affinity matrix and further adopt the label propagation technique to refine pseudo-labels for the target domain. Moreover, they also use batch normalization adaptation and self-supervised learning methods, including data augmentation, to improve their algorithm. The experimental results on several SFDA benchmarks verify the effectiveness of the proposed method.

**Strengths:**

- The paper is well-organized and easy to follow.

- The idea of introducing label propagation in SFDA is interesting. However, some recent works [1] also mentioned this research topic, which reduces the novelty of this paper.

- The proposed method is concise, and the experimental results are relatively promising.

 [1] Chunwei Wu, Guitao Cao, Yan Li, Xidong Xi, Wenming Cao, Hong Wang, “Chaos to Order: A Label Propagation Perspective on Source-Free Domain Adaptation.” ACM Multimedia 2023

**Weaknesses:**

- My main concern with this paper is about the novelty. The proposed method has two main contributions: the label propagation technique and the self-training framework. Recently, the self-training and data augmentation methods have been widely used in SFDA research. And some recent papers also adopted the label propagation in SFDA. [1,2] There lacks some necessary discussions compared with the state-of-the-art works.

- I have some questions about the experimental part. Please refer to the Question section.


[1] Chunwei Wu, Guitao Cao, Yan Li, Xidong Xi, Wenming Cao, Hong Wang, “Chaos to Order: A Label Propagation Perspective on Source-Free Domain Adaptation.” ACM Multimedia 2023

[2] Nazmul Karim, Niluthpol Chowdhury Mithun, Abhinav Rajvanshi, Han-pang Chiu, Supun Samarasekera, and Nazanin Rahnavard, “C-sfda: A curriculum learning aided self-training framework for efficient source free domain adaptation.” CVPR, 2023

**Questions:**

I have some questions about the experimental part.

- To build your affinity matrix while using the historical models, should you save all previous affinity matrices and also the neighbor banks? Is this method of storage consuming and effective?

- By observing the curves in Fig 4 left panel, does this mean the utility of the fused affinity matrix will be continuously decreasing during the adaptation process?

- Also, in Figure 4, right panel, why, at epoch 0 (the initial step), are the ratios of two affinities not equal?

---

> ### Author Response · Authors · 2023-11-17
> **Rebuttal**
>
> Thank you for your time and suggestive comments. We have addressed the questions below.
>
> **W1.** Thank you for pointing out the need for a more detailed discussion of related works. We have added the discussion of relevant works in Section A5.
>
> In addition, we agree that the self-training framework is a prevalent choice in SFDA and many existing SFDA methods adopt a self-training framework. We emphasize that our distinctive contribution lies in the innovative combination of Label Propagation with the self-training framework, resulting in enhanced adaptation performance.
>
> Concerning CtO[1], it used a local clustering approach instead of label propagation (even though the title contains label propagation). This approach shares similarities with NRC and AaD as they all aim to adjust model predictions based on their immediate neighbors, emphasizing local structure. However, our method employs label propagation, which exploits both global and local structures of the target samples. Another significant contribution involves integrating the historical model into the label propagation process. While leveraging historical models is not novel in semi-supervised learning, direct ensembling of historical models for pseudo-label refinement in SFDA often yields suboptimal results due to the lack of adaptation of historical models to the target domain. Motivated by the observation that similar samples form clusters in the feature space using both current and historical models, we leverage a combination of both to construct a more accurate affinity matrix. This integration is essential for enhancing the effectiveness of label propagation and, consequently, improving overall performance.
>
> **Q1.** For historical models, there is no necessity to store the neighbor banks; only the affinity matrix needs to be preserved. The affinity matrix itself is highly sparse, occupying approximately 6MB for VisDA and 2MB for Office-Home. In an effort to reduce both storage consumption and input/output operations (IO), we conducted experiments by storing the affinity matrix every two epochs and fusing the matrices from time step $\lfloor(t-m)/2\rfloor * 2$, resulting in equivalent outcomes.
>
> **Q2.** Indeed, this trend is expected. As the training advances, the model converges gradually, leading to minimal disparities between the current and historical models. Once label propagation and model fine-tuning reach an equilibrium, neither label propagation nor historical affinity fusion will further enhance the model.
>
> **Q3.** Sorry for the confusion. This discrepancy arises from the design of our implementation of historical affinity fusion (HAF). For time step $t$, where $t<m$,  we fuse the affinity matrix with the initial affinity matrix obtained at epoch 0. Thus, for epoch 0, the affinity matrix is fused with itself. The figure plots the ratio/percentage of connected samples, weighted by similarity, sharing the same label. Consequently, the fused affinity matrix at epoch 0 assigns higher weights to neighbors with greater similarity compared to the vanilla affinity.

---

> > ### Comment · Reviewer_113e · 2023-11-22
> >
> > Thanks for the authors' response. I have also read the other reviewers' comments and the rebuttals. I agree that by combining some current techniques, the proposed HAF-based label propagation method could help with source-free domain adaptation problems. However, I still have some concerns about the novelty of this paper, and I will keep my original score.

---

### Official Review · Reviewer_CVkN · 2023-11-01

**Soundness:** 4 excellent
**Presentation:** 4 excellent
**Contribution:** 2 fair
**Rating:** 6
**Confidence:** 4

**Summary:**

This paper proposes a new approach for source-free domain adaptation called Enhanced Label Propagation through Affinity Matrix Fusion. This approach is designed to enhance the quality of pseudo-labels in the target domain by synergizing the predictions of an old model with the current model. To achieve this, the method constructs an affinity matrix and employs label propagation techniques to refine pseudo-labels. These refined labels are then used to calculate gradients from an alternative augmentation perspective. Additionally, the approach incorporates a prediction diversity loss and leverages BN adaptation techniques. Experimental evaluations are conducted on popular datasets such as Office-31, Office-Home, and VisDA-C, yielding results that are comparable to state-of-the-art methods.

**Strengths:**

1. The paper is well written.
2. As stated by the authors, label propagation technology was originally employed for making predictions on unlabeled data in a semi-supervised setting. However, this paper introduces a novel application by using it to refine pseudo-labels.

**Weaknesses:**

1. This proposed method combines several previously established techniques, such as label propagation, consistent loss with two views, BN adaptation, and prediction diversity loss. It lacks novelty in terms of its technical components. It seems the sole innovation lies in applying label propagation technology for refining pseudo labels additionally using historical model outputs.
2. The experimental results show only a marginal advantage on Office-31 and Office-Home datasets.

**Questions:**

1. Why exclusively employ the previous model h_{t-m} instead of a fused model (such as \sum_{i=1}^{m} h_{t-i}) for retrieving historical information?
2. Is there a particular rationale behind the choice of the negative dot product loss as opposed to other commonly used loss functions? I couldn't find any ablation results in the paper.

---

> ### Author Response · Authors · 2023-11-17
> **Rebuttal**
>
> Thank you for your time and suggestive comments.  I have addressed the questions below.
>
> **W1.** Thank you for your valuable suggestions. We have opted to utilize only the previous model $h_{t-m}$ for both efficiency and effectiveness considerations. The model from the more immediate history tends to be more similar to the current model.
> We have conducted experiments that fuse the current model with two historical models, including the model at time $t-m$ and the model at time $t-\lfloor m/2 \rfloor$. The corresponding results are presented in the table below, from which we did not observe a significant improvement by integrating more historical models.
>
> | Method                           |VisDA     |
> | --------                            | -------    |
> | Ours w/o HAF                 |    88.9   |
> | Ours HAF with $h_{t-m}$|   89.6   |
> | Ours HAF with $h_{t-m}$ + $h_{t-\lfloor m/2 \rfloor}$ |  89.5   |
>
>
>
> **W2.** The use of the negative dot product is inspired by previous works NRC and AaD, where the dot product loss has proven effective in the context of SFDA. Additional rationale for the use of the negative dot product loss is detailed in Appendix A.2. Particularly, it is a combination of squared error loss and a component that encourages prediction discriminability. Below, we have compared the average accuracy of our approach with both negative dot product loss and cross-entropy loss across all datasets, which proves the effectiveness of negative dot product loss in the context of SFDA.
>
> | Method            |VisDA     | Office-Home| Office31 |
> | --------             | -------    | -------           |-------     |
> | Ours                |    89.6    | 73.5              | 90.6       |
> | Ours w/ CE loss |  86.8   | 69.8              | 88.7       |

---

### Official Review · Reviewer_j3cS · 2023-11-01

**Soundness:** 3 good
**Presentation:** 2 fair
**Contribution:** 2 fair
**Rating:** 5
**Confidence:** 5

**Summary:**

This paper focuses on source-free domain adaptation and proposes to utilize an affinity matrix derived from current and historical models during the label propagation process. The proposed method takes advantage of different snapshots of the model to obtain a more accurate representation of the underlying graph structure, significantly enhancing the efficacy of label propagation and resulting in more refined pseudo-labels. Extensive experiments show that the proposed method achieves superior domain adaptation performance, which does not require source domains.

**Strengths:**

-a. It incorporates label propagation to leverage both local and global relationships between the instances in the target domain for source-free domain adaptation.
- b. It investigates the critical role of data augmentation as a fundamental component in the self-training-based SFDA framework.
- c. The experiments are extensive, and the performance of the proposed method is promising in source-free domain adaptation scenario.

**Weaknesses:**

1. The proposed method seems a bit similar to HCL (Huang et al., 2021) which also exploits current and historical models for source-free domain adaptation. It would be better to discuss and compare the differences, pros and cons of HCL and the proposed method.
2. The proposed label propagation method seems similar the method of “Temporal Assembling for semi-supervised learning”. It would be better to discuss and compare the differences, pros and cons of this method and the proposed method.
3. The ablation study is not clear. Why the baseline in Table 4 outperforms the source model by a large margin? Do all the other methods in other tables use the same baseline as in Table 4?

**Questions:**

see Weaknesses

---

> ### Author Response · Authors · 2023-11-17
> **Rebuttal**
>
> Thank you for your comments. We have addressed the questions below.
>
> **W1.** Thank you for your valuable feedback. We have already cited these two works in our original submission. In addition, we have added a detailed discussion of these relevant works in Section A.5. While acknowledging the similarities between HCL and our proposed method, it's essential to highlight two fundamental distinctions: 1) HCL relies on the agreement between current and historical models to access the confidence of the pseudo-label without involving a specific refinement process.  In contrast, our approach adopts label propagation for pseudo-label refinement, an important step that significantly improves the accuracy of the adaptation process. 2) Our method goes beyond a simple reliance on historical models; we specifically leverage them to construct a more precise nearest-neighbor graph. This strategic incorporation is integral to the effectiveness of label propagation and underscores the nuanced ways in which historical models contribute to the method's performance.
>
> **W2.** The approach outlined in "Temporal Ensembling for Semi-Supervised Learning (Laine, S. and Aila, T., 2016)" ensembles predictions from current and historical models and uses the ensembled prediction as the target or pseudo-labels. However, in the context of SFDA, where the model is not fully adapted to the target domain, directly using the ensembled source model prediction as pseudo-labels is not viable. In this scenario, label propagation becomes crucial for refining pseudo-labels. As articulated in the main paper, we refine the pseudo-labels using the equation  $Q^{\ast} = (I - \alpha S) ^{-1} P$,  where  $S$ is the normalized affinity matrix,  $Q^{\ast}$ and $P$ represent the final refined pseudo-label matrix and the raw prediction matrix respectively.
>
> To explore various ways of integrating historical models, we compare our baseline model without Historical Affinity Fusion with the following two approaches: (1) Following the methodology of Laine and Aila (2016), we use the moving average of predictions from the current and historical models as entries for $P$ (Baseline+MAP). (2) We utilize both the current and historical models to build the affinity matrix (Baseline+HAF), which is our adopted final solution. Intriguingly, compared to our baseline, approach (1) may even degrade model performance, while approach (2) consistently improves it. We conjecture that in the source-free domain adaptation scenario, historical models are inadequately adapted to the target domain, leading to noisy predictions and potential confirmation bias, making direct use of the ensembled predictions for $P$ less effective. On the other hand, historical models prove beneficial for improving the affinity matrix, as similar samples tend to form cohesive clusters in the feature space using either source or target models.
> | Method            |Avg Acc. on VisDA|
> | --------             | -------    |
> | Baseline           |    88.9   |
> | Baseline+MAP |   88.7    |
> | Baseline+HAF  |   89.6    |
>
> **W3.** Sorry for the confusion. The baseline in Table 4 represents our baseline approach with vanilla label propagation for pseudo-label refinement,  without incorporating historical affinity fusion or employing strong augmentation in the self-training stage (outlined in the caption of Table 4). It's expected that our baseline model should improve over the source model by a large margin.

---

### Meta-Review · Area_Chair_FDQZ · 2023-12-08

**Metareview:**

Thanks for your submission to ICLR.

This paper explores a label propagation-based technique for source-free domain adaptation, and applies it to some standard benchmark data sets.

Reviewers were on the fence with this paper, ultimately giving two weak accept and two weak reject scores.  In the reviews, the biggest concern, raised by multiple reviewers, was in the novelty of the approach, which borrowed several existing ideas but put them together.  For instance, label propagation and the use of historical models have both been done.  Further, there were multiple questions raised in terms of the ablation study of the paper.  The authors did a good job responding to these concerns but some concerns lingered, and the reviewers were somewhat split on the paper.

I took a look at the paper as well.  I am also on the fence here, but tend to lean toward not accepting the paper.  I agree that the paper's contribution is somewhat on the weak side.  Further, I would have like to have seen some additional results on challenging data sets such as DomainNet.  In the end I think this paper falls just under the bar for ICLR.

**Justification For Why Not Higher Score:**

This was a borderline paper, and could have gone either way but there are some fundamental issues of novelty here.  Further, two of the four reviewers are on the side of rejecting the paper, so it's difficult to advocate for accept in this case.

**Justification For Why Not Lower Score:**

N/A

---

### Decision · Program_Chairs · 2024-01-16

Reject